# Memory T cells targeting oncogenic mutations detected in peripheral blood of epithelial cancer patients

Gal Cafri[1], Rami Yossef[1], Anna Pasetto[1], Drew C. Deniger [1], Yong-Chen Lu [1], Maria Parkhurst[1], Jared J. Gartner[1], Li Jia[1], Satyajit Ray[1], Lien T. Ngo[1], Mohammad Jafferji[1], Abraham Sachs [1], Todd Prickett [1], Paul F. Robbins[1] & Steven A. Rosenberg[1]

T cells targeting shared oncogenic mutations can induce durable tumor regression in epithelial cancer patients. Such T cells can be detected in tumor infiltrating lymphocytes, but whether such cells can be detected in the peripheral blood of patients with the common metastatic epithelial cancer patients is unknown. Using a highly sensitive in vitro stimulation and cell enrichment of peripheral memory T cells from six metastatic cancer patients, we identified and isolated CD4$^+$, and CD8$^+$ memory T cells targeting the mutated KRAS$^{G12D}$ and KRAS$^{G12V}$ variants, respectively, in three patients. In an additional two metastatic colon cancer patients, we detected CD8$^+$ neoantigen-specific cells targeting the mutated SMAD5 and MUC4 proteins. Therefore, memory T cells targeting unique as well as shared somatic mutations can be detected in the peripheral blood of epithelial cancer patients and can potentially be used for the development of effective personalized T cell-based cancer immunotherapy across multiple patients.

[1] Surgery Branch, National Cancer Institute, National Institutes of Health, Bethesda, MD 20892, USA. These authors contributed equally: Gal Cafri, Rami Yossef. Correspondence and requests for materials should be addressed to S.A.R. (email: sar@nih.gov)

Tumors express proteins harboring unique mutations that are absent from normal tissue. Some of these mutated proteins can trigger specific T-cell responses and therefore can potentially be recognized as neoantigens. Recent studies have demonstrated that tumor-infiltrating lymphocytes (TILs) are enriched with neoantigen-specific T cells[1–6] and that adoptive cell therapy (ACT) using neoantigen-specific TIL can sometimes lead to durable tumor regression[4,7–9]. However, owing to tumor heterogeneity, targeted neoantigen(s) can be expressed in some, but not all, tumor cells, which may limit ACT efficacy. Therefore, targeting common oncogenic mutations that are more likely to be expressed in all tumor cells and are essential for tumor survival represents a more promising approach. We have recently shown that ACT using autologous TILs targeting the HLA-C*08:02 restricted $KRAS^{G12D}$ epitope could lead to tumor regression in a patient with metastatic colon cancer[7]. However, T cells targeting common oncogenic mutations are rarely found in TILs and new, non-invasive, approaches for the identification and isolation of such cells or their T-cell receptors from TIL or circulating lymphocytes is needed.

Two major approaches have been used recently to enrich neoantigen-reactive cells from the peripheral blood of melanoma patients: PD-1-positive (PD-1+) enrichment of CD8+ T cells[10] and tetramer isolation[1]. However, isolation of neoantigen-specific cells from the blood of patients with the common metastatic epithelial cancers has been much more challenging. In general, the average number of mutations in common epithelial cancers is lower than in melanoma and may lead to a limited repertoire of neoantigen-reactive TILs[11]. The low frequency of neoantigen-reactive T cells in the periphery requires highly sensitive isolation methods. In addition, unlike melanoma, establishing autologous cell lines from excised epithelial tumors is challenging with low success rates. The absence of autologous lines to validate tumor recognition by enriched T cells and the need to avoid raising de novo recognition against irrelevant antigens suggests that new approaches should focus on T-cell populations that are more likely to be clinically relevant. Although the naive T-cell (TN) repertoire is highly polyclonal and antigen inexperienced, the memory repertoire represents cells that have already been stimulated by their cognate antigens and more likely arose following infection or malignancy. Thus, the limited antigen-experienced repertoire of memory cells is ideal for in vitro stimulation (IVS)-based enrichment and isolation methods from circulating T cells. The cells or their receptors identified using such approaches are likely to arise from antigens that are efficiently processed and presented in the tumor microenvironment or its draining lymph nodes. Here, we developed a highly sensitive IVS approach starting with preexisting memory T cells and used our approach to detect and isolate T cells targeting unique as well as shared somatic mutations in the KRAS oncogene.

## Results

### Neoantigen specific TCR sequences are restricted to peripheral memory T cells

Recently, Gros et al. and others[10,12,13] have shown that sorting PD-1+ TILs and peripheral blood enriches for tumor-reactive CD8+ cells in patients with melanoma. To determine whether the neoantigen-reactive T cells are present in the memory compartment of epithelial cancer patients, we initially tested the phenotype of PD-1+ T cells in the peripheral blood of four metastatic gastrointestinal cancer patients (patients 4217, 4254, 4257, and 4252). Analysis of all four samples revealed that the majority of CD3+PD-1+ T cells were central memory T cells (TCM) or effector memory T cells (TEM) (Fig. 1a, b), whereas no TN cells were shown to be PD-1-positive. Interestingly, the majority of the terminally differentiated effector memory

(TEMRA) cells did not express PD-1. To further address whether neoantigen-specific T cells were enriched in the memory T-cell subset, we retrospectively sorted and performed TCR-Vβ deep-sequencing of TN, TCM, TEM, and TEMRA cells from peripheral blood lymphocytes (PBL) of six epithelial cancer patients with known neoantigen-reactive TCRs that were previously found in their TILs (ref. [14], and data not published). Comparison of the TCR-Vβ sequences obtained from TILs with the sequences of the matched PBL subsets shows that neoantigen-reactive TCRs were detected only in memory cells (TCM, TEM, and TEMRA) but not in TN cells (Fig. 1c, and Supplementary Figure 1A). The frequency of the neoantigen-reactive T cells in the blood was low, ranging from 0.02 to 0.0007% of each T-cell subpopulation. Further analysis of the productive clonality based on TCR-Vβ sequences revealed a clear hierarchy of clonality among the different T-cell populations (Fig. 1d), thus demonstrating that TEM and TEMRA memory populations are significantly more clonal than TN cells.

### Isolation of T cells targeting unique mutations

To test the feasibility of using memory T cells to isolate neoantigen-reactive T cells, we developed a novel IVS method (Supplementary Figure 2) and retrospectively tested our approach using PBLs from two metastatic colon cancer patients (patient 4213 and 4217, Supplementary Figure 3 and Supplementary data 1). Patient 4213 was screened in our laboratory for the presence of neoantigen-specific TILs for potential ACT, as described earlier[2]. In brief, using whole-exome and RNA sequencing we identified somatic mutations that were present in two metastatic tumors derived from patient 4213. Two neoantigens were identified in the initial screen, SMAD5[P268inPKH], and DDX1[S281F] (Supplementary Figure 4, Supplementary data 2). Here, we sorted memory (TCM, TEM, and TEMRA), TN and bulk PBL cells from peripheral blood mononuclear cell (PBMC) of Pt. 4213 (Supplementary Figure 1A) and co-cultured them with dendritic cells (DCs) loaded with RNA encoding tandem minigenes (TMGs) for 14 days in the presence of IL-21, IL-7, and IL-15. Each TMG comprises a string of RNA minigenes encoding identified mutations flanked on each side by 12 wild-type amino acids from the parent protein. After 14 days, memory and TN cells were re-stimulated with DCs loaded with all TMGs and sorted based on CD8+, CD4+, and expression of the T-cell activation marker 41BB to enrich for neoantigen-reactive T cells (Fig. 2a). The number of activated CD8+ T cells sorted was low, 962 from memory, and 1100 from TN (Fig. 2a), possibly owing to the low frequency of neoantigen-specific cells, as shown in Fig. 1c. Cells were then expanded and screened against all 13 TMGs to test for neoantigen recognition. Memory and naive CD8+ T cells were reactive against TMG-8 and TMG-6, respectively, whereas bulk PBL did not recognize any TMGs (Fig. 2b). To identify the specific neoantigens in TMG-8 and TMG-6, we co-cultured the enriched memory and TN cells with autologous DCs that were individually pulsed with the mutated peptides encoded by TMG-8 or TMG-6 (Supplementary Table 1). TMG-8-reactive CD8+ memory T cells recognized the SMAD5[P268inPKH] mutation (Fig. 2c), whereas the TMG-6-reactive CD8+ TN cells did not recognize any single peptide from TMG-6 (Fig. 2d). The SMAD5[P268inPKH] reactive memory CD8+ cells were further tested for the recognition of WT and mutated long peptides, the predicted minimal epitope and a full-length SMAD5 RNA corresponding to the mutated and WT protein sequences. As shown in Fig. 2e the SMAD5-reactive cells recognized the mutated and not the WT LP, minimal epitope and full-length SMAD5 RNA. We fluorescence-activated cell sorting (FACS)-purified the SMAD5[P268inPKH]-reactive cells from Pt.4213 TIL and memory cells and performed single-cell RT-PCR (scPCR)

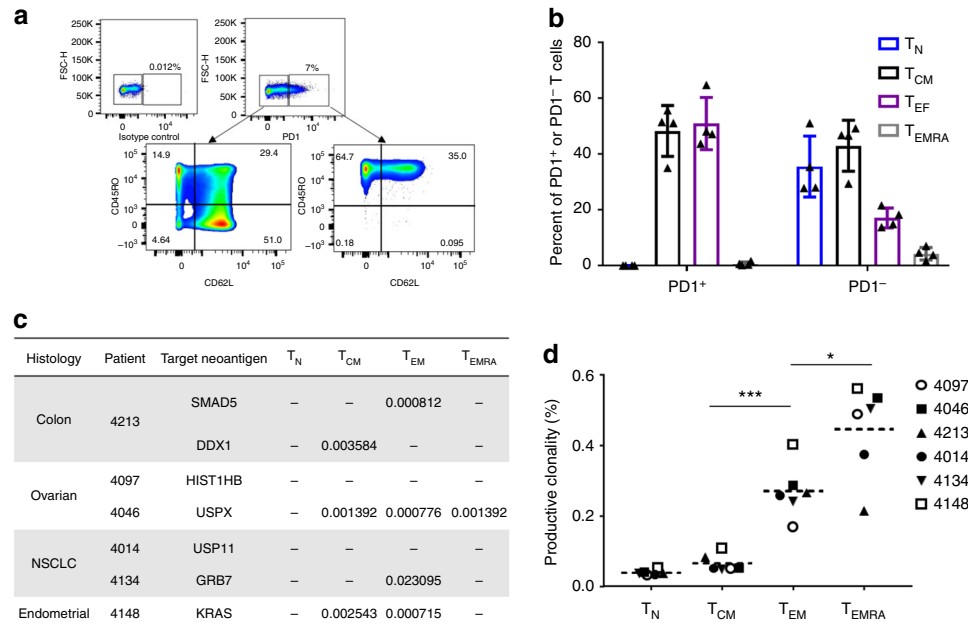

**Fig. 1** Identification of neoantigen-specific T cells in PBMC of epithelial cancer patients. **a** A representative data from one of four patients showing the gating strategy used for the phenotypic analysis of PD-1-positive (PD-1⁺) peripheral T cells. **b** PBMC isolated from four epithelial cancer patients were thawed, rested overnight in a cytokine-free T-cell medium and stained with antibodies against CD3, CD45RO, CD62L, and PD-1. PD-1⁺ and PD-1-negative (PD-1⁻) cells were analyzed for the surface expression of the T-cell memory markers, CD45RO and CD62L. Error bars indicate the mean standard deviation between all four patients samples. **c** PBMC isolated from six epithelial cancer patients were thawed, rested overnight in a cytokine-free T-cell medium and stained with antibodies against CD3, CD45RO, CD62L, CCR7, and CD45RA. Cells were sorted based on their memory subset, washed once with PBS, frozen at −80 °C and send for TCR-Vβ sequencing. Numbers represent the percent of specific TCR-Vβ sequence in each T-cell population. A minus sign (−) was used If no sequence was identified. **d** Analysis of the productive clonality of TCR-Vβ sequences of each T-cell population isolated in **c**. (Paired *t*-test, *** *P* < 0.001, * *P* < 0.05) dashed line, mean

to identify their TCR-Vβ and TCR-Vα sequences[15]. Analysis of the TCR sequences revealed that the SMAD5^P268inPKH-reactive TIL- and IVS-derived memory cells share matching TCR-Vβ and TCR-Vα sequences (Supplementary Tables 2 and 3). Genetically engineered autologous peripheral blood T cells expressing the SMAD5^P268inPKH-reactive TCR (Supplementary Table 3) conferred that the selective reactivity of the TCR against the mutated SMAD5^P268inPKH peptide is restricted by HLA*B08:01 (Fig. 2f, g). Thus, using IVS of memory T cells can lead to the identification and enrichment of neoantigen-reactive T cells.

**Isolation of T cells targeting shared oncogenic mutations.** Owing to tumor heterogeneity and differential expression of nonsynonymous mutations in cancer cells, targeting mutated driver mutations, that appear to be homogenously expressed in cancer cells (Supplementary Table 4), can target cancer cells more efficiently. Therefore, we sought to determine whether T cells targeting shared oncogenic mutations can be isolated using our approach. We decided to focus on hotspot mutations in the protooncogene *KRAS*, which are shared at high frequencies across multiple cancers histologies[16,17]. T cells targeting mutated human *KRAS* hotspot mutations were previously reported in human TILs[3], vaccinated mice[18], human PBMC[19–21], and in two clinical trials involving vaccination with mutated KRAS peptides[22,23].

To test the use of this approach across multiple epithelial cancer types, we performed IVS of memory, $T_N$, and bulk PBL, from a metastatic endometrial cancer patient (pt.4148), using a mixture of KRAS^G12V, KRAS^G12C, and KRAS^G12D 24mer peptides, followed by 41BB enrichment. Exome and RNA analysis of Pt.4148 tumor showed expression only of the *KRAS^G12V*

mutation (Supplementary data 3 and Supplementary Figure 5). No TILs reactive against the *KRAS^G12V* were identified in the initial screen performed in our laboratory (Supplementary Figure 6). To identify T cells targeting mutated *KRAS*, enriched memory, $T_N$, and PBL cells were co-cultured with autologous DCs pulsed with the individual KRAS^G12 mutated long peptides. As shown in Fig. 3a, memory CD8 T cells showed selective reactivity against KRAS^G12V LP. $T_N$ and bulk PBL were not reactive against any of the KRAS peptides (Supplementary Figure 7). As Pt.4148 expressed the HLA-A*11:01, which is predicted to bind *KRAS^G12V* 9mer (Supplementary Table 5), we presumed that staining the cells with A*11-9mer tetramers could address whether HLA-A*11:01 is the correct restriction element. Indeed, HLA-A*11:01-9mer tetramers bound 15.9% of the memory CD8 cells in the culture (Fig. 3b). Sorting the tetramer-positive cells and sequencing their TCR revealed single TCRα and TCRβ chains (Supplementary Table 3). To further test the TCR, we synthesized, cloned and retrovirally transduced the TRAV and TRBV into allogeneic PBL_S, as previously described[3,10]. To evaluate the specificity and the potency of the TCR we co-cultured the TCR-transduced PBLs with cancer cell lines harboring KRAS^G12V mutations with or without transfection with HLA-A*11:01 (Fig. 3c) or with autologous DCs pulsed with a serial dilution of the mutated 9mer and wild-type peptides (Fig. 3d). The results show that the isolated TCR selectively recognize the KRAS G12V mutation presented on HLA-A*11:01. Owing to the high prevalence of KRAS^G12V expression across cancers and HLA-A*11:01 allele frequencies in selected populations (14% in US Caucasians and 23% in Asian-Americans)[24] this TCR can potentially be used as off-the-shelf reagent to treat thousands of relevant cancer patients per year.

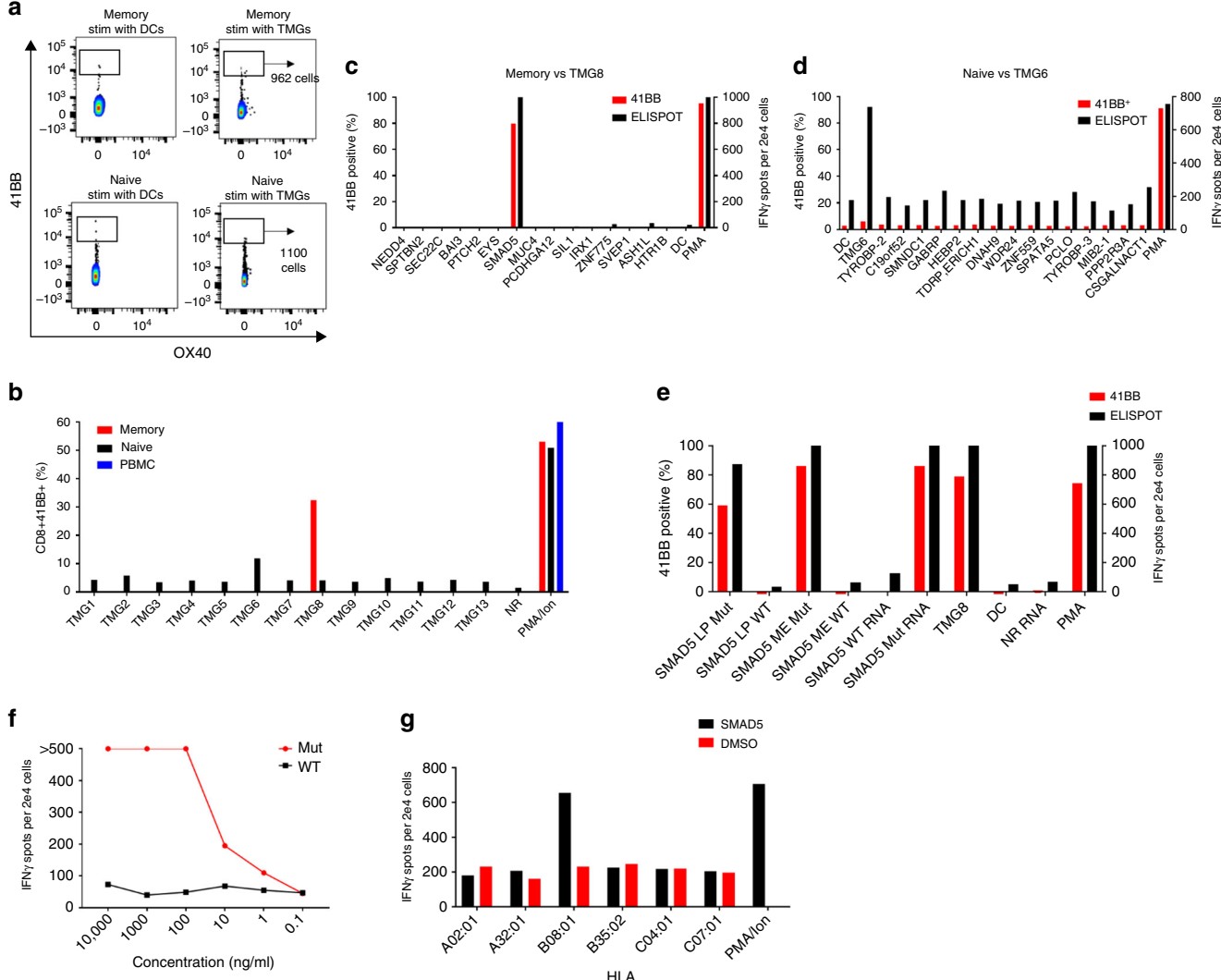

**Fig. 2** Identification of T cells specific for mut SMAD5. **a** Memory and naive CD8 cells were co-cultured with autologous DCs transfected with 13 TMGs harboring 201 mutated 25mer sequences for 18 h. Activated T Cells were stained for CD3, CD8, 41BB, and OX40 and sorted based on 41BB and OX40 expression to enrich for neoantigen-reactive cells. **b** Memory, $T_N$ and PBMC were co-cultured with autologous DCs transfected with TMGs 1–13 for 18 h, stained with CD3, CD8, and 41BB and analyzed for surface expression of 41BB as a marker for T-cell activation. **c** Memory CD8 cells isolated in A were co-cultured for 18 h with autologous DCs that were individually pulsed with the mutated peptides encoded by TMG-8 and tested either by flow cytometry for 41BB expression or IFNγ-secretion using ELISPOT assay. **d** Naive CD8 cells isolated in **a** were co-cultured for 18 h with autologous DCs that were individually pulsed with the mutated peptides encoded by TMG-6 and tested either by flow cytometry for 41BB expression or IFNγ-secretion using ELISPOT assay. **e** Memory CD8 cells isolated in **a** were co-cultured for 18 h with autologous DCs that were loaded with WT or Mut SMAD5 LP, predicted minimal epitope (ME) and full-length WT and mutated SMAD5 RNA. Cells were tested for antigen recognition by flow cytometry for 41BB expression or IFNγ-secretion using ELISPOT assay. **f** TCR-transduced PBLs were co-cultured with DCs pulsed with a serial dilution of SMAD5 Mutated or WT peptides. **g** COS7 cells were transfected with patient's class I HLA and co-cultured with TCR-transduced cells. Reactivity was determined by ELISPOT for IFNγ

Next, we sought to apply our approach to identify and isolate TCRs targeting additional mutated KRAS neoepitopes. To that end, we stimulated T-cell subsets from PBL of a metastatic rectal cancer patient (Pt.4171). Exome and RNA analysis of Pt.4171 tumor showed expression only of the KRAS<sup>G12D</sup> mutation (Supplementary data 4). The PBL were stimulated using KRAS<sup>G12D</sup> full-length RNA or 24mer peptide following ten days of IVS, the T-cell subsets were re-stimulated, and FACS sorted based on T-cell activation markers, and further expanded, as described above. Enriched cells were further tested for recognition of KRAS<sup>G12D</sup>. Both memory and bulk sorted, CD4 cells showed specific recognition of KRAS<sup>G12D</sup> 24mer peptide-pulsed on autologous DCs (Fig. 4a). To isolate the reactive TCR, we performed an 18 h co-culture of these subsets with autologous

DCs pulsed with the mutated peptide and sorted T cells that upregulated activation markers (OX40[+], 41BB[+], or OX40[+]41BB[+] double positive) into a 96-well PCR plate for scPCR[15]. The subsets shared one TCR (TCR1), however, in the memory subset the sequencing revealed two additional TCRs (Fig. 4b). The TCRs were constructed, cloned and retrovirally transduced into allogeneic PBLs, as described above. Only TCR1, which was present in both sorted subsets, showed specific reactivity against the G12D peptide (Fig. 4c). Next, we evaluated the specificity and avidity of the TCR against DCs pulsed with a serial dilution of mutated and wild-type KRAS peptides (Fig. 4d) and determined that TCR recognition is restricted by HLA-DRB1*08:01 (Fig. 4e). In summary, we employed our IVS approach on T-cell PBL subsets from six metastatic cancer patients that harbored *KRAS*

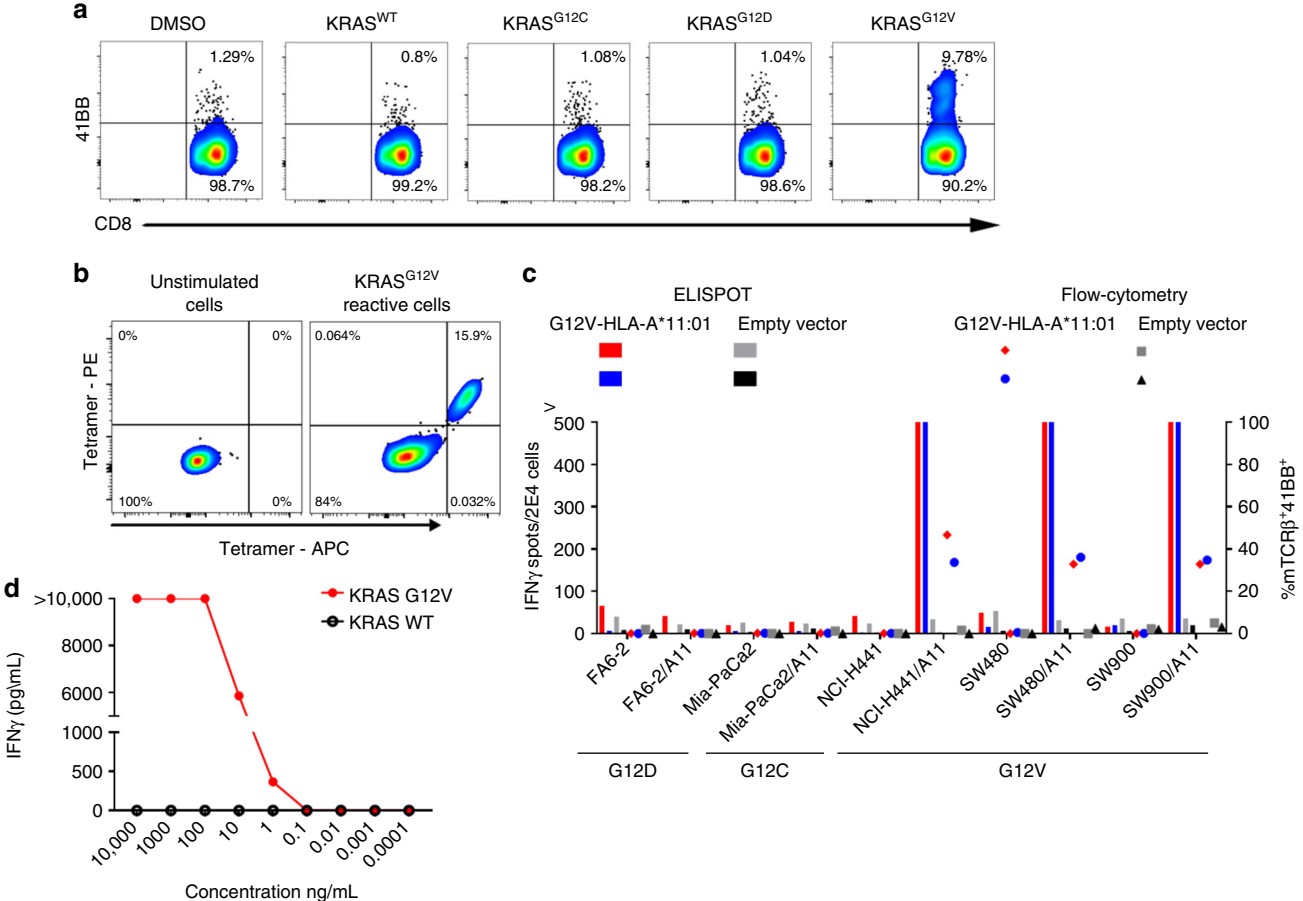

**Fig. 3** Identification, isolation, and analysis of KRAS$^{G12V}$-reactive TCR. **a** 41BB-enriched CD8 memory T cells were expanded, and their reactivity was tested against autologous DCs pulsed with the indicated peptides. **b** 41BB-enriched CD8 memory T cells are stained with KRAS$^{G12V}$ 9mer tetramers; unstimulated CD8 cells were used as control (left). **c** 41BB upregulation and ELISPOT IFNγ secretion assay of TCR-transduced allogeneic T cells. T cells co-cultured with cell lines naturally expressing G12 mutations ± HLA-A11 transduction. **d** IFNγ secretion of TCR-transduced cells against autologous DCs pulsed with the indicated concentrations of mutated and WT 9mers. A representative of at least three experiments

mutation in their tumor. In three we were able to isolate TCRs targeting KRAS mutation from their memory cells (Pt.4148, Pt. 4171, Pt.4238 presented in Fig. 3, Fig. 4, Supplementary Figure 8 and Supplementary data 5, respectively). In Pt.4171 PBLs we isolated the reactive TCR from bulk CD4 as well; however, none of the reactive TCRs were detected in the naïve subset (Supplementary Table 6).

## Discussion

Employing this novel approach enabled us to identify and isolate neoantigen-reactive T cells that were present at very low frequencies in the circulation of metastatic cancer patients. Although all current IVS methods are performed on bulk or naive PBLs[25,26], our method enables the isolation of T cells or TCRs from the limited repertoire of memory precursors, which is of high relevance for personalized treatments. We identified a total of two reactivities against neoantigens, both of them also detected in TIL from these patients, and three against KRAS from a total of five patients. In all cases except one (Pt.4171), the specific cells were detected in the memory cells and not in T$_N$ or bulk PBL (Supplementary Table 6). In one case (pt.4213) the T$_N$ showed recognition of TMG-6 that has not been explained. The efficiency of isolating reactive cells from memory is possibly owing to the clonal nature of this population and the elimination of false positive reactivities arising from de novo activation of cells from

T$_N$. Previously we showed that ACT with TIL targeting KRAS mutation led to substantial tumor regression in a metastatic cancer patient[7] and as KRAS mutations are highly prevalent among different cancer histologies (Supplementary Table 7), isolating TCRs targeting such mutations can be of high clinical relevance to a substantial portion of cancer patients. From a total of six patients screened using our IVS method, we were able to isolate and clone eight different memory derived TCRs targeting mutated KRAS from three patients. The TCRs can be further assessed for clinical use with gene-engineered PBMCs. The ability to identify and isolate T cells or T-cell receptors from the peripheral blood of patients with the common epithelial cancers could potentially be used to obtain T cells targeting those tumors without any need for tumor harvest to obtain TIL. By applying exome and RNA sequencing on tissue obtained during primary tumor resection, cells can be isolated from patient PBL and potentially be used for treatment upon relapse. Our results provide strong evidence that peripheral blood-memory T cells can serve as a reliable source of neoantigen-reactive T cells, and the TCRs isolated from reactive cells can be used for personalized treatment. In the case of mutated shared oncogenes, off-the-shelf reagents could be used.

## Methods

**IVS of naive and memory T cells**. Apheresis samples were thawed, washed, set to 5–10e6 cells/ml with AIM-V media (Life Technologies) and $1.75–2 \times 10^8$ viable

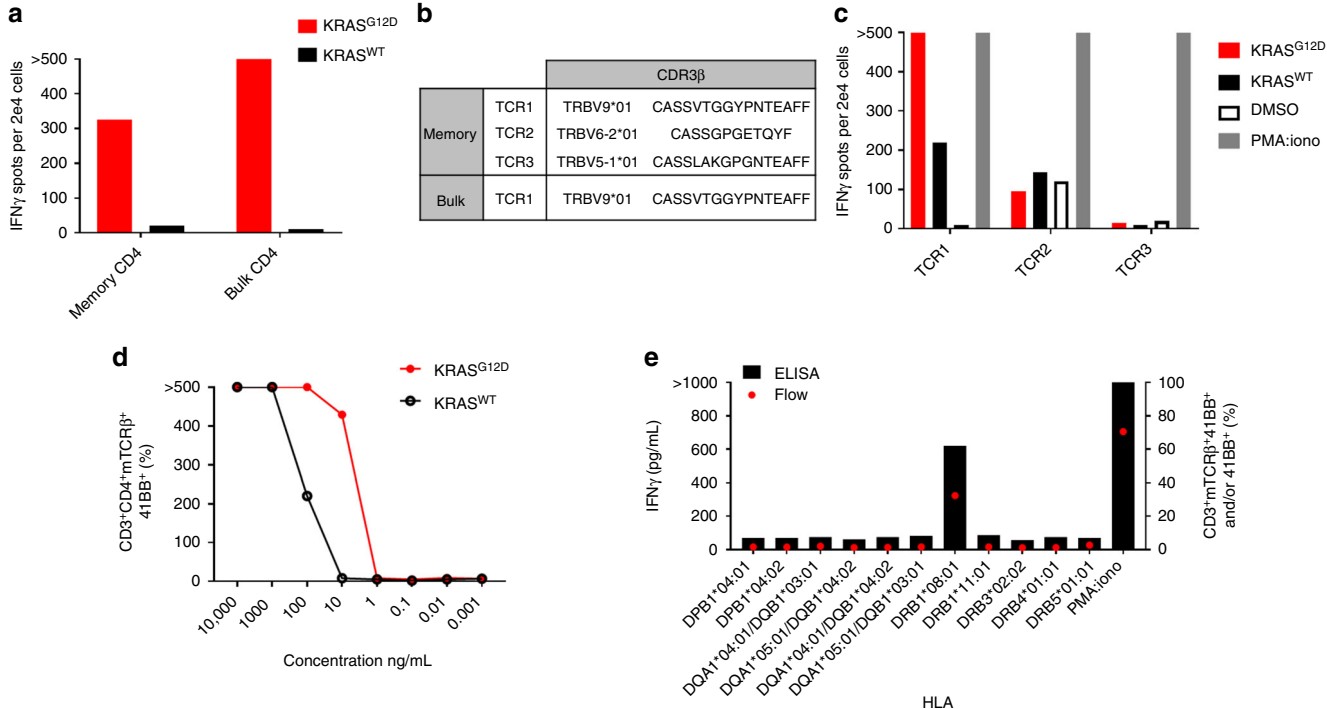

**Fig. 4** Isolation of HLA-DRB1*08:01-restricted TCR-targeting KRAS$^{G12D}$. **a** IFNγ-ELISPOT assay of 41BB$^+$ and/or OX40$^+$ enriched CD4 subsets co-cultured with autologous DCs pulsed with mutated and wild-type KRAS peptides. **b** TRBV sequences of 41BB$^+$-sorted T cells following co-culture with DCs pulsed with KRAS$^{G12D}$ peptide. **c–e** Retrovirally transduced allogeneic PBLs expressing the indicated TCRs were further tested. **c** Transduced PBLs were co-cultured with autologous DCs pulsed with either KRAS$^{G12D}$ or KRAS$^{WT}$. **d** TCR1-transduced PBLs were co-cultured with DCs pulsed with a serial dilution of G12D or WT peptides. **e** COS7 cells were transfected with patient's class II HLA and co-cultured with TCR1-transduced cells. Reactivity was determined by IFNγ ELISA and the upregulation of 41BB surface marker. **c–e** are representative of at least two experiments

cells were incubated in T175 flasks (Corning Inc.) at 37ºC. After 2 h, the flasks were washed 2–3 times vigorously with phosphate-buffered saline (PBS) to collect non-adherent cells for T-cell sorting. For the adherent cells, 30 ml DC media were added, comprised of RPMI containing 5% human serum, 100 U/ml penicillin and 100 μg/ml streptomycin, 2 mM L-glutamine, 800 IU/ml GM-CSF (Leukine) and 200 U/ml IL-4 (Peprotech), and cells were incubated at 37ºC, 5% CO$_2$. On day 4 or 5, cells were harvested and freshly used or frozen for further use. DCs were seeded into low-attachment 12 or six-well plates for peptide loading or TMG transfection. For peptide loading, DCs were loaded with 10–15 μg/μl peptide or peptide pools for 2–12 h. For TMG, RNA transfection lipofectamine or electroporation were used and the cells were incubated for 8-12 h prior to IVS 1. DCs were harvested by washing with PBS and incubated for 5 minutes in 5 ml 0.9 mM ethylenediamine-tetraacetic acid-phosphate-buffered saline. DCs were washed with DC medium and resuspend at a concentration of 5e5/ml.

Non-adherent cells were spun, resuspended in 50/50 medium comprising 1:1 mix of RPMI-1640 with L-glutamine (Lonza) and AIM-V (Gibco) supplemented with 100 U/ml penicillin, 100 μg/ml streptomycin, 12.5 mM HEPES, and 5% human serum, and rested overnight at 37ºC, 5% CO$_2$. Cells were then harvested, washed twice with cold sorting buffer (PBS$^{-/-}$, 1%FCS, 0.5 mM EDTA), resuspend in sorting buffer at 1e6/100 μl and stained with fluorescent-conjugated primary antibodies against CD3, CD8, CD4, CD62L, CD45RO, CD45RA, and CCR7.

T cells from sorted populations were collected, count and spin. T cells were resuspended in CTL medium at a concentration of 2e6/ml. IL-21, 60 ng/ml were added to the T-cell fraction. (This will result in a final concentration of 30 ng/ml after the addition of the DCs). DCs were mixed with T cells at a 1:1 (vol/vol) ratio (resulting in a 4:1 T-cell:DC ratio, 1e$^6$ T cells /2.5e$^5$ DCs). In total, 500 μl of the cell mix were transferred into individual wells of a 48-well plate. Cells were incubated at 37 ºC for 72 h.

First feeding—Check cells under the microscope and calculate the required amount of medium to give 0.5 ml per well. In all, 500 μl of warm CTL medium containing 60 ng/ml of IL-21 and 3000 IU/ml of IL-2 (referring to the final concentration in the culture medium). were added to each well and incubate at 37 ºC for 72 h.

Second feeding—1 ml of warm CTL medium containing 60 ng/ml of IL-21 and 3000 IU/ml of IL-2 was added to each well of a 12-well plate. Cells and medium from each well of the old plate were transferred to the new 12-well plate. Cells were incubated for 48 h.

Third feeding—2 ml of warm CTL medium containing 60 ng/ml of IL-21 and 3000 IU/ml of IL-2 (referring to the final concentration in the culture medium)

were added to each well of a six-well plate. Cells and medium from each individual well of the old 12-well plate were transferred to individual wells of the new six-well plate. Cells were incubated for 72 h. From this step cells can be kept in CTL medium containing 3000 IU/ml IL-2 and expanded until the second IVS and 41BB/OX40 enrichment.

***For T$_N$ in patients 4148, 4171, and 4238, instead of IL-21 and IL-2, 5 ng/ml of IL-15 and IL-7 were used in the second feeding and 10 ng/ml in the third feeding.

Immature DCs were thawed resuspended in DC medium + cytokines and transferred into low-attachment 12- or 6-well plates for peptide loading. For peptide loading: DCs were loaded with 10–15 μg/μl peptide or peptide pools for 2–12 h. Peptide-loaded DCs were washed and mixed at 1:4 ratio (DC:T) with the appropriate T-cell population (usually 4–8e6 T cells were used for each stimulation). Cells were plated into 12- or 6-well plates in 3 ml CTL medium and incubated for 18–24 h. cells were harvested, washed with sorting buffer (PBS, 1% fetal calf serum (FCS), 0.5 mM EDTA) and stained with Anti-human CD8, Anti-human CD3, Anti-human 41BB, Anti-human OX40 and Anti-human CD4. Samples were washed with sorting buffer and the CD3 + CD4 + 41BB + OX40 +/ CD3 + CD8 + 41BB + OX40 + cells (both double and single positive cells) were sorted. After sorting cells were kept in CTL medium supplemented with 1500 IU/ml IL-2 (if the cell number is lower than 1e5, cells were REPed and seeded in T25 flasks). Cells were grown for 11–16 days and screened against individual peptides, peptides pools or TMG's.

Monocyte-derived, immature dendritic cells were generated using the plastic adherence method[3,10]. In brief, autologous apheresis samples were thawed, washed, set to 5–10e6 cells/ml with AIM-V media (Life Technologies) and then incubated at ~1e6 cells/cm$^2$ in an appropriate size tissue culture flask and incubated at 37 ºC, 5% CO$_2$. After 120 min, non-adherent cells were collected, and the flasks were vigorously washed with PBS, and then adherent cells were incubated with RPMI (Life technologies) containing 5% human serum 100 U/ml penicillin and 100 μg/ml streptomycin, 2 mM L-glutamine, 800 IU/ml GM-CSF and 800 U/mL IL-4 (Peprotech). On day 4–7, fresh DCs were collected. Fresh or freeze/thawed DCs were used in experiments on day 4–5 after initial stimulation.

Genomic DNA and Total RNA was purified using the QIAGEN AllPrep DNA/RNA (cat #80204) for patients 4213, 4148, 4238, and 4171 fresh tumor (FrTu) and matched normal apheresis samples following manufacturer's suggestions. Whole-exome library construction and exon capture of ~ 20,000 coding genes was prepared using Agilent Technologies SureSelectXT Target Enrichment System (cat# 5190-8646) for paired-end libraries coupled with Human All Exon V6 RNA

bait (cat# 5190-8863) (Agilent Technologies, Santa Clara, CA, USA). WES libraries were subsequently sequenced on a NextSeq 500/550 desktop sequencer (Illumina, San Diego, CA, USA). The library was prepped using gDNA (3 μg) isolated from the fresh tumor tissue following the manufacturer's protocol. Paired-end sequencing was done with an Illumina High-output flow cell kit (300 cycles) (cat# FC-404-2004) using v2 of reagent/flow cell kit. Furthermore, RNA-Seq libraries were prepared using 2 μg of Total RNA with the Illumina TruSeq RNA Stranded library prep kit following the manufacturer's protocol. RNA-Seq libraries were paired-end sequenced on our NextSeq 500/550 desktop sequencer (Illumina, San Diego, CA, USA) again using the same mechanism described above to generate 25 + million paired-end reads.

**Sequence alignment, processing, and variant calling**. Output from the sequencer was de-multiplexed and converted to fastq format using illumina's bcl2fastq program. Reads were trimmed for quality and to remove and adapter sequence with Trimmomatic software[27]. Once trimmed, Exome reads are aligned to hg19 genome using novoalign from novocraft[28] to create initial starting bams. RNA-seq reads are aligned to hg19 using STAR two pass alignment process[29]. Both RNA-seq and Exome bam files are preprocessed according to the GATK best practices protocol. Exome SNVs are called using Strelka, Somatic Sniper, Varscan2 and Mutect. Insertions and deletions (In/Dels) are called using Strelka and Varscan2. For neoantigens arise from single nucleotide variants (SNVs) cutoff criteria for evaluation of a variant is Tumor and normal coverage of 10 or greater, Tumor variant read count of 4 or greater, tumor variant frequency of 7% or greater and 2 or greater callers calling that variant. For neoantigens arise from In/Dels the criteria is the same except there is no caller criteria. RNA variants are called with Varscan with no cutoffs. Somatic variants are annotated using Annovar against three separate reference databases (Refgene, Ensembl, UCSC). All variants that pass cutoff criteria and those found in COSMIC regardless of cutoff criteria have neoepitopes generated using an in-house python script. This script produces 25 mers with 12 aa flanking the mutation on either side where possible. In the event, it is not possible due to the mutation be located closer than 12 aa from the beginning or end of a transcript, the maximum number of aa that can flank are used. For In/Del mutations, the corresponding change is made to the cDNA sequence, and then 12 aa before the mutation (where possible) are extracted as well as all amino acids beyond the mutation up until the 1st stop codon is encountered. If no stop codon is encountered, the neoepitope will encompass all sequence up to the end of the cDNA transcript.

**Construction of tandem minigene and in vitro transcription**. For TMGs construction[2,13], each non-synonymous variant identified a minigene was constructed encoding the mutant amino acid flanked by 12 amino acids of the wild-type sequence. TMGs were cloned into pcRNA2SL using *Eco*RI and *Bam*HI. Following linearization of the constructs, phenol–chloroform extraction was performed and precipitated the DNA with sodium acetate and ethanol. Next, we used 1 μg of linearized DNA to generate in vitro-transcribed RNA using the mMessage mMachine T7 Ultra kit (Life Technologies) as instructed by the manufacturer. RNA was precipitated using LiCl₂ and resuspended at 1 μg/μl.

**Peptide pulsing and RNA liposomal transfection**. Peptides were made in-house or purchased from GenScript or peptides & elephants. In brief, autologous or allogeneic DCs were harvested, washed and resuspended at 0.5–1e6 cells/mL concentration in DC media supplemented with 800 IU/mL GM-CSF and 800 U/mL IL-4. Next, cells were incubated with peptides for 2–12 h at 37ºC, 5% CO₂. Prior to co-culture DCs were collected, washed twice with PBS, and resuspended in 50/50 media and then used for co-culture assays.

RNA transfection was done with mMessengerMAX reagent (Life Technologies) according to manufacturer's instructions. In brief, DCs were harvested, washed, and resuspended at 0.75–1e6 cells/mL and then 0.5 mL were seeded in low-attachment 24-well plate at 0.75–1e6 cells/mL in DC media supplemented with IL-4 and GM-CSF for 4–12 h at 37ºC, 5% CO₂. Following, the mMessengerMAX-Opti-MEM mixture was incubated for 10 minutes at room temperature (RT). In total, 200–500 ng/well RNAs were diluted in Opti-MEM and mixed with mMessengerMAX-Opti-MEM in 1:1 ratio and incubated for additional 5 minutes in RT, then the mixture was added to cells and incubated 8–20 h at 37ºC, 5% CO₂.

**Co-culture assays: IFN-γ ELISPOT, ELISA assays and flow cytometry for activation markers CD134 and CD137 staining**. When DCs were used as target cells, 3e4–1e5 cells/well were used in 96-well plates. When cells lines were used as target cells, 2e4–5e4 cells/well were used in 96-well plates. 5e3–2e4 1e42e4 cells/well effector T cells were used in 96-well plates. All co-cultures were performed in 50/50 media in the absence of exogenously added cytokines. Phorbol 12-myristate 13-acetate: ionomycin mixture (eBioscience) was used as a positive control.

In HLA-blocking assays target cells were incubated with 20–50 μg/mL blocking antibodies for 2 h at 37ºC, 5% CO₂ and then effector cells were added and incubated for 12-18 h.

IFN-γ ELISPOT assays were performed in MultiScreen-IP filter plates (EMD Millipore). Each plate was pretreated with 50 μl 35% or 70% ethanol/well for < 2 min, washed 4 × with Ultra-pure water (Quality Biological) and then coated with 10 μg/ml of an IFN-γ capture antibody (100 μl/well, clone: 1-D1K, Mabtech, diluted in PBS) overnight at 4 °C. Anti-CD3 antibody (clone: OKT3, 1–10 μg/ml) was added to the positive control wells. At the day of co-culture, each plate was washed 5 × with PBS and blocked with complete medium without IL-2 for at least 30 min at room temperature. After overnight co-culture (18–24 h), the cells were harvested and transferred into a round-bottom 96-well plate for flow cytometry staining and analysis. Each ELISPOT plate was washed 5 × with PBS containing 0.05% Tween 20 (MP Biomedicals) and then incubated for 2 h with 1 μg/ml, 0.22 μm-filtered anti-human-IFN-γ detection antibody (clone: 7-B6-1, Mabtech, 100 μl/well, diluted in PBS + 0.5% fetal bovine serum (FBS)). Next, each plate was washed 5 × with PBS and incubated for 1 h with streptavidin–ALP (Mabtech, 100 μl/well, 1:3,000 diluted with PBS + 0.5% FBS), followed by three washes with ddH₂O and development with 0.45-μm-filtered BCIP–NBT substrate solution (KPL, 100 μl/well) for 5–10 min. The reaction was stopped by rinsing thoroughly with cold tap water. After they completely dried, each ELISPOT plate was scanned and counted using an ImmunoSpot plate reader and associated software (Cellular Technologies). The cells from the co-culture were harvested from ELISPOT wells prior to IFNγ spots development, stained for flow cytometry and surface expression of CD134 and CD137 was assessed using BD FACSCantoI, BD FACSCantoII, or BD LSR Fortessa. All flow cytometry data were analyzed using FlowJo software (TreeStar Inc).

**Single-cell sorting and single-cell RT-PCR**. Single-cell reactive T cells were sorted into 96-wells plate containing RT-PCR buffer based on activation markers (CD134, CD137) or tetramer staining using FACSAria instrument (BD Biosciences). TCR sequences from the sorted single cells were obtained by a series of 2 nested PCR reactions[15]. Multiplex PCR with multiple Vα and Vβ region primers (Supplementary data 6) and one primer for Cα and Cβ regions each was performed using the One-Step RT-PCR kit (Qiagen). The RT-PCR reaction was performed accordingly to manufacturer's instructions using the following cycling conditions: 50 °C 15 min; 95 °C 2 min; 95 °C 15 s, 60 °C 4 min × 18 cycles; 4 °C. For the second amplification reaction, 4 μl from the first RT-PCR product were used as a template in total 25 μl PCR mix using HotStarTaq DNA polymerase (Qiagen) and multiple internally nested Vα and Vβ region primers and 1 internally nested primer for Cα and Cβ regions each (final concentration of each primer is 0.6 μM). The cycling conditions were 95 °C 15 min; 94 °C 30 s, 50 °C 30 s, 72 °C 1 min × 50 cycles; 72 °C 10 min; 4 °C. The PCR products were purified and sequenced by Sanger sequencing method with an internally nested Cα and Cβ regions primers by Beckmann Coulter.

**TCR survey and deep sequencing**. TCR-Vβ deep sequencing was performed by immunoSEQ, Adaptive Biotechnologies (Seattle, WA) on genomic DNA isolated from peripheral blood T cells, and frozen tumor tissues. T-cell numbers in sequenced samples ranged from ~2 × 10⁴ cells to 1 × 10⁶. TRB clonality and productivity were analyzed using immunoSEQ Analyzer 3.0 (Seattle, WA). Only productive TCR rearrangements were used in the calculations of TCR frequencies.

**TCR cloning, retrovirus production, and transduction of T cells**. For TCR cloning and transduction of T cells[3,13], TRAV-J-encoding sequences were fused to mouse TCRβ constant chain, and TRBV-D-J–encoding sequences were fused to mouse TCRα constant chain[30]. Mouse constant chains were modified to improve TCRαβ pairing[30]. The full-length TRB and TRA chains were separated by a furin SGSG P2A linker. TCR construct was cloned into a pMSGV1 retroviral vector.

For transduction, autologous, or allogeneic apheresis samples were thawed and set to 2e6 cells/ml in T-cell media, which consists of a 50/50 mixture of RPMI and AIM-V media supplemented with 5% in-house human serum, 10 μg/ml gentamicin (CellGro), 100 U/ml penicillin and 100 μg/ml streptomycin, and 2 mM L-glutamine (all from Life Technologies). In total, 2e6 cells/ml were stimulated in a 24-well plate with 50 ng/ml soluble OKT3 (Miltenyi Biotec) and 300 IU/ml IL-2 (Chiron) for 2 days prior to retroviral transduction. Retroviral supernatants were generated in HEK-293GP packaging line[3,15]. In brief, pMSGV1 plasmid encoding mutation-specific TCR (2 μg/well) and the envelope-encoding plasmid RD114 (0.75 μg/well) were co-transfected with 1e6 239GP cells per well of a six-well poly-D-lysine–coated plates using Lipofectamine 2000 (Life Technologies). Retroviral supernatants were collected at 42-48 h after transfection, diluted 1:1 with DMEM media, and then centrifuged onto Retronectin-coated (10 μg/ml, Takara), non–tissue culture-treated six-well plates at 2000 g for 2 h at 32 °C. Stimulated T cells (2e6 per well, at 0.5e6 cells/ml in IL-2 containing T-cell media) were then spun onto the retrovirus plates for 10 min at 300–350 g. Stimulated T cells were transduced overnight, removed from the plates and further cultured in rIL-2-containing T-cell media. GFP and mock transduction controls were included in transduction experiments. Cells were typically assayed 10–14 days post retroviral transduction.

### Antibodies

*Patient informed consent*. All patient samples were obtained in the course of a National Cancer Institute Institutional Review Board-approved clinical trial. Patients provided informed consent.

The following titrated anti-human antibodies were used for cell surface staining: CCR7- BB515 (cat. 565869, 1:12.5 dilution) or FITC (cat. 561271, 1:12.5 dilution), CD45RO-PE-Cy7 (cat. 560608, 1:7 dilution) or APC (cat.559865, 1:7 dilution), CD45RA-APC (cat. 561210, 1:7 dilution) or BD Horizon™ V450 (cat. 560362, 1:7 dilution), CD62L-PE (cat. 555544, 1:7 dilution), CD4-PE (cat. 566679, 1:100 dilution), FITC (cat. 56561005, 1:100 dilution), BV605 (cat. 562659, 1:100 dilution), CD3- AF700 (cat. 561027, 1:100 dilution) or APC-H7 (cat. 560176, 1:100 dilution), CD4-FITC, PE, PE-Cy7, APC-H7 (clone: SK3), CD8-PE-Cy7 (cat. 335787, 1:100 dilution), OX40-PE-Cy7 (cat. 563663, 1:7 dilution) or FITC (cat. 555837, 1:7 dilution), 41BB-APC (cat. 550890, 1:7 dilution). All antibodies were from BD Biosciences. For MHC blocking assays, the following antibodies were used: pan-class-I (clone: W6/32, produced from hybridoma and used at 1:20 dilution), pan-class-II (BD pharmingen, cat. 555556, 1:20 dilution), HLA-DR (clone: HB55, produced from hybridoma and used at 1:20 dilution), HLA-DP (Leinco, cat.H127, used at 15 μg/ml), and HLA-DQ (BD pharmingen, cat. 555562, 1:20 dilution). For cells stimulation, purified anti-CD3 was used (Miltenyi Biotech, cat.130-093-387, used at 30 ng/ml).

**Reporting summary**. Further information on experimental design is available in the Nature Research Reporting Summary linked to this article.

## Data availability

The exome and RNA sequencing data for all patients have been deposited in the sequence read archive at NCBI under the accession code PRJNA507557. Each sample include the patient name, number of the resected metastatic lesion that was sequenced (1Met for example) and the sequencing output (exome or RNA-seq). Normal tissue sequencing data is marked as N (for example 4217N-exome). Patient 4217 exome and RNA sequencing data have been deposit under the accession codes 4217-3Met-exome, 4217-2Met-exome, 4217-1Met-exome, 4217-3Met-RNA-seq, 4217-2Met-RNA-seq, 4217-1Met-RNA-seq, 4217N-exome. Patient 4213 exome and RNA sequencing data have been deposit under the accession codes 4213-3Met-RNA-seq, 4213-2Met-RNA-seq, 4213N-exome, 4213-2Met-exome, and 4213-2Met-exome. Patient 4238 exome and RNA sequencing data have been deposit under the accession codes 4238Met-exome, 4238N-exome, and 4238Met-RNA-seq. Patient 4148 exome and RNA sequencing data have been deposit under the accession codes 4148-2Met-RNA-seq, 4148-1Met-RNA-seq, 4148-1Met-exome, 4148N-exome, and 4148-2Met-exome. Patient 4171 exome and RNA sequencing data have been deposit under the accession codes 4171Met-RNA-seq, 4171N-exome, and 4171Met-exome.

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

## Acknowledgements

We thank K. Hanada for providing KRAS-mutated cell lines, A. Mixon and S. Farid for flow cytometry support, and other members of the Surgery Branch for helpful discussions and technical support. We thank the medical arts service at the NIH for figure editing. A patent applications related to KRAS^G12V (U.S. application no. 62/594,244) was filled. This research was supported by the Center for Cancer Research intramural research program of the National Cancer Institute.

## Author contributions

G.C. developed the IVS method and performed the experiments regarding the identification of T cells targeting unique mutations. R.Y. performed the experiments regarding the identification of T cells targeting shared oncogenes. A.P. performed the TCR single-cell sequencing. D.C.D and M.P. contributed to the development of the IVS method and for some of the experiments. Y.C.L performed single-cell sequencing for the identification of the G12V-specific TCR. J.J.G and L.J. performed all bioinformatic analysis. S.R., L. T.N, M.J, and A.S performed experiments. T.P. performed the DNA and RNA sequencing. P.F.R contributed to the development of the IVS method and experiments design. S. A.R contributed to the development of the IVS method, experiments design, and overlooked the project. All authors discussed the results. G.C., R.Y., and S.A.R. wrote the manuscript with input from all other authors.

## Additional information

**Competing interests:** The authors declare no competing interests.

