## [Peer Review File · Nature Communications]

Reviewers' Comments:

Reviewer #1:

Remarks to the Author:

Immunotherapy involving adoptive cell transfer has much potential for cancer treatment, but a major limitation has been the difficulty in isolating truly tumor-specific T cells. The work described in this manuscript represents a highly significant and novel advance in this field by demonstrating the isolation of tumor-specific T cells from peripheral blood of patients. The authors use a novel means of isolating and expanding these T cells involving sorting and expansion of memory T cells. The authors used sophisticated technology involving whole exome sequencing to identify mutations in tumor cells, and in vitro screening to identify which mutations induced T cell responses. They demonstrate that neoantigen-reactive T cells could be isolated from the blood of a significant proportion of patients.

This is an extensive study, across multiple patients with different malignancies, involving the isolation and characterization of T cells reactive with mutated proteins.

The study also describes the cloning of DNA encoding TCRs, which were used to validate their tumor reactivity, and highlighted the potential for generating large numbers of tumor-specific T cells for patient treatment. Importantly, TCRs targeting common shared oncogenic mutations could be isolated in a proportion of patients, which highlights the potential for generating reagents applicable across many patients. This is an important study of potentially far-reaching significance.

Minor comments:

- (1) The manuscript would be easier to follow if the Results section was divided into subsections with appropriate titles, and the final paragraph was subdivided and labeled "Discussion".
- (2) The legend to Fig 4 has a typo error – two panels C are listed.
- (3) Y-axis of Fig 4D is mislabeled.

-Mike Kershaw, Peter MacCallum Cancer Center

Reviewer #2:

Remarks to the Author:

The paper of Cafri et al. addresses the use of PBMC from cancer patients to obtain shared mutation-specific T cells (KRAS) for future personalized treatment. It shows an isolation procedure reporting how "regular" neo-antigen specific T cells from the blood can be obtained by focusing on the memory T cell pool, and then how KRAS-specific CD4 or CD8 T cells could be obtained in 3 of the 6 patients with epithelial cancer. The outlook of using the isolated TCR for off-the-shelf therapies is of interest, but not the topic of this paper.

Although the experiments have been well performed and the data clearly sustain their conclusions, there are several issues to be addressed:

P5, line 112: Since PD-1 is only expressed after TCR-mediated activation of T cells it is not very informative that PD-1 was non detected on truly naive T cells. More importantly, however, the authors state that neoantigen-reactive were only found in memory cells. This is a bit too bald and should be rephrased to "detected only" as clearly this is a frequency/detection problem.

P5, line 120: the authors should explain their definition of "highly sensitive IVS method". Furthermore, it is a stimulation method and does not refer to a detection, which can be sensitive.

P5-7, lines 119-156 describe basically the validation of their IVS method, including several figures and tables. Although this all is a nice piece of work it is not addressing the key question of this article as it describes the work on non-shared mutations (regular neoantigens). To my surprise

this method, using TMG-transfected DC, was not used for the isolation of KRAS-specific T cells. For these patients either peptide-pulsed DC or full length RNA-transfected DC were used. A formal comparison of these methods is missing. Hence, the value of this large part of the manuscript is low.

P7, lines 158-159 the authors state "targeting mutated driver mutations", which may be rephrased. Can they provide evidence for their statement that these are homogenously expressed in cancer cells? For instance by NGS amplicon sequencing of their patients cancers.

P7, line 175: data not shown, please provide as supplemental data, as these are the first data using PBL and naive T cells with the peptide IVS approach, which was not shown in the first validation part of the IVS method. There are whole series of reports on stimulation of TAA and mutation-specific T cells from naive T cells using peptide stimulation approaches, hence it will be important to see what happened with the method used here.

P8, line 201 it is not clear if the authors sorted either OX40+ and 4-1BB+ or OX40+4-1BB double positive cells here. Please rephrase for the reader.

Overall, the authors are not the first to show that neo-antigen specific T cells, in particular KRAS-specific T cells are present in PBMC of patients and can be stimulated. This is already acknowledged by them referencing to refs 1 and 10. But it is also shown by others, for example, Labarriere CII 2008, Verdegaal CII 2011 and Nature 2016 also showed that patient PBMC have are a good source for tumor-specific T cells, including neoantigen-specific T cells. The idea to target shared somatic mutations is also not new and have been pioneered by the group of Gaudernack for shared KRAS and TGF β mutations. This group, already reported the detection of PBMC-derived memory CD4+ T cells recognizing KRAS mutation specifically, in two different HLA class II molecules, and that such responses are not found in healthy donors or in patients not likely to have a tumor with such a mutation (Gedde-dahl Int Immunol. 1992 / EJI 1994), as well as reported the detection and isolation of CD4 and CD8 T cells to KRAS mutated peptide found in PBMC of a cancer patient which specifically recognized tumor cells with that mutation (Fossum Int J Cancer / CII 1995). Which made them conclude already at that time that such common mutations may be a target for treatment purposes.

Reviewer #3:

Remarks to the Author:

The manuscript details out an in-vitro stimulation approach to isolate and enrich cancer associated antigen specific T-cells from peripheral blood of epithelial cancer patients. Using memory compartment of T-cell repertoire, the authors demonstrate identification and expansion of very low frequency cancer associated T-cells (shared and personalized neoantigen) across three different cancer types. The data support claim of isolating low-frequency T-cells is feasible with this approach only from memory precursor and not from bulk peripheral blood lymphocytes in all but one patient sample. The alternative approach used in this study provides better detection of cancer associated antigens thus could potentially beneficial for clinical applications.

Specific comments:

1. More details should be added on the improvement of sensitivity, looking in the memory compartment rather than PBMCs. Data is included for 2 patients only on this matter, and it is important for the overall conclusion of the paper. E.g. a table listing the detection different PBMC vs memory compartment could be useful.
2. Figure 2: Neoantigen reactive T-cells identified for patient 4213 (colon cancer) has been reported by this group previously, thus the current approach doesn't identify any new neoantigen reactive T-cells from an unknown sample. Should be emphasized
3. Figure 2: For better clarity, authors might consider swapping Figure 2B with 2A. This would establish first that no recognition was observed when bulk PBMCs were stimulated with mutated

long peptides.

4. More information should be added on the frequency of the mutations across patient populations. Again a table format could be useful. Arguing the value of these neoepitopes as a shared target would be supported by such information.

Response to reviewers

Reviewers' comments:

Reviewer #1: Expert in immunotherapy
(Remarks to the Author):

Immunotherapy involving adoptive cell transfer has much potential for cancer treatment, but a major limitation has been the difficulty in isolating truly tumor-specific T cells. The work described in this manuscript represents a highly significant and novel advance in this field by demonstrating the isolation of tumor-specific T cells from peripheral blood of patients. The authors use a novel means of isolating and expanding these T cells involving sorting and expansion of memory T cells. The authors used sophisticated technology involving whole exome sequencing to identify mutations in tumor cells, and in vitro screening to identify which mutations induced T cell responses. They demonstrate that neoantigen-reactive T cells could be isolated from the blood of a significant proportion of patients. This is an extensive study, across multiple patients with different malignancies, involving the isolation and characterization of T cells reactive with mutated proteins.

The study also describes the cloning of DNA encoding TCRs, which were used to validate their tumor reactivity, and highlighted the potential for generating large numbers of tumor-specific T cells for patient treatment. Importantly, TCRs targeting common shared oncogenic mutations could be isolated in a proportion of patients, which highlights the potential for generating reagents applicable across many patients. This is an important study of potentially far-reaching significance.

Minor comments:

(1) The manuscript would be easier to follow if the Results section was divided into subsections with appropriate titles, and the final paragraph was subdivided and labeled "Discussion."

Response: We agree with the reviewer and therefore divided the results section into two different subsections. The final paragraph is now labeled as Discussion.

(2) The legend to Fig 4 has a typo error – two panels C are listed.

Response: Thank you for noticing. The legend was changed

(3) Y-axis of Fig 4D is mislabeled.

Response: The label was added

-Mike Kershaw, Peter MacCallum Cancer Center

**Reviewer # 2: Expert in T cells
(Remarks to the Author):**

The paper of Cafri et al. addresses the use of PBMC from cancer patients to obtain shared mutation-specific T cells (KRAS) for future personalized treatment. It shows an isolation procedure reporting how “regular” neo-antigen specific T cells from the blood can be obtained by focusing on the memory T cell pool, and then how KRAS-specific CD4 or CD8 T cells could be obtained in 3 of the 6 patients with epithelial cancer. The outlook of using the isolated TCR for off-the-shelf therapies is of interest, but not the topic of this paper.

Although the experiments have been well performed and the data clearly sustain their conclusions, there are several issues to be addressed:

P5, line 112: Since PD-1 is only expressed after TCR-mediated activation of T cells it is not very informative that PD-1 was non detected on truly naive T cells. More importantly, however, the authors state that neoantigen-reactive were only found in memory cells. This is a bit too bald and should be rephrased to “ detected only” as clearly this is a frequency/detection problem.

Response: We agree with the reviewer and the sentence was rephrased as follow:

“ ...neoantigen-reactive TCRs were **detected** only in memory cells”

P5, line 120: the authors should explain their definition of “ highly sensitive IVS method”. Furthermore, it is a stimulation method and does not refer to detection, which can be sensitive.

Response: The sentence was rephrased to: “we developed a novel IVS method..”

P5-7, lines 119-156 describe basically the validation of their IVS method, including several figures and tables. Although this all is a nice piece of work, it is not addressing the key question of this article as it describes the work on non-shared mutations (regular neoantigens). To my surprise this method, using TMG-transfected DC, was not used for the isolation of KRAS-specific T cells. For these patients either peptide-pulsed DC or full-length RNA-transfected DC was used. A formal comparison of these methods is missing. Hence, the value of this large part of the manuscript is low.

Response: We agree with the reviewer, it is important to evaluate the potential of sensitizing T cells with TMGs for the enrichment of reactive cells using our method. However, the results presented in the paper were obtained from experiments done before we constructed a plasmid with mutated *KRAS* TMGs. Furthermore, a substantial proportion of our neoantigen specific TIL in the common epithelial cancers are CD4 rather than CD8 cells and using peptides for the stimulation can cover both.

P7, lines 158-159 the authors state “ targeting mutated driver mutations” , which may be rephrased. Can they provide evidence for their statement that these are homogenously expressed in cancer cells? For instance by NGS amplicon sequencing of their patients cancers.

Response: We added Table.S5 that show the homogeneity and the RNA expression of *KRAS* mutations in 21 patients admitted for treatment in the Surgery Branch. Only patients that had more than one metastasis excised and sequenced were included in the analysis (n=21). A total of 48 out of 50 metastases showed RNA expression of the mutated gene. Also, we rephrased the following sentence:

“appear to be homogenously expressed in cancer cells (Table S5)”

P7, line 175: data not shown, please provide as supplemental data, as these are the first data using PBL and naive T cells with the peptide IVS approach, which was not shown in the first validation part of the IVS method. There are whole series of reports on stimulation of TAA and mutation-specific T cells from naive T cells using peptide stimulation approaches, hence it will be important to see what happened with the method used here.

Response: We added Fig. S7 showing upregulation of 4-1BB in CD8 bulk and naïve PBLs following co-culture with autologous DCs pulsed with KRAS 24mer peptides. These results are from the same experiment shown in Fig.3.

P8, line 201 it is not clear if the authors sorted either OX40+ and 4-1BB+ or OX40+4-1BB double positive cells here. Please rephrase for the reader.

Response: The sentence was rephrased to the following:

“To isolate the reactive TCR, we performed an 18 hrs co-culture of these subsets with autologous DCs pulsed with the mutated peptide and sorted T cells that upregulated activation markers (OX40⁺, 4-1BB⁺ or OX40⁺4-1BB⁺ double positive) into a 96-well PCR plate for scPCR”

Overall, the authors are not the first to show that neo-antigen specific T cells, in particular KRAS-specific T cells are present in PBMC of patients and can be stimulated. This is already acknowledged by them referencing to refs 1 and 10. But it is also shown by others, for example, Labarriere CII 2008, Verdegaal CII 2011 and Nature 2016 also showed that patient PBMC have are a good source for tumor-specific T cells, including neoantigen-specific T cells. The idea to target shared somatic mutations is also not new and have been pioneered by the group of Gaudernack for shared KRAS and TGFB mutations. This group, already reported the detection of PBMC-derived memory CD4⁺ T cells recognizing KRAS mutation specifically, in two different HLA class II molecules, and that such responses are not found in healthy donors or in patients not likely to have a tumor with such a mutation (Gedde-dahl Int Immunol. 1992 / EJI 1994), as well as reported the detection and isolation of CD4 and CD8 T cells to KRAS mutated peptide found in PBMC of a cancer patient which specifically recognized tumor cells with that mutation (Fossum Int J Cancer / CII 1995). Which made them conclude already at that time that such common mutations may be a target for treatment purposes.

Response: We agree with the reviewer that the rational of using IVS to detect and target driver mutations is not new. However, here we are trying to emphasis the feasibility of our new method in detecting and isolating low-frequency T cell precursors targeting KRAS mutations from the blood. Regarding the comment related to unique neoantigenes (Labarriere CII 2008, Verdegaal CII 2011 and Nature 2016), they are all describing the identification of tumor specific PBMC from the blood of melanoma patients. The focus of this paper is on the common epithelial cancers, not melanoma. We updated the references list to include the papers described by reviewer 2.

Reviewer #3: Expert in Neoantigens
(Remarks to the Author):

The manuscript details out an in-vitro stimulation approach to isolate and enrich cancer associated antigen specific T-cells from peripheral blood of epithelial cancer patients. Using memory compartment of T-cell repertoire, the authors demonstrate identification and expansion of very low frequency cancer associated T-cells (shared and personalized neoantigen) across three different cancer types. The data support claim of isolating low-frequency T-cells is feasible with this approach only from memory precursor and not from bulk peripheral blood lymphocytes in all but one patient sample. The alternative approach used in this study provides better detection of cancer associated antigens thus could potentially be beneficial for clinical applications.

Specific comments:

1. More details should be added on the improvement of sensitivity, looking in the memory compartment rather than PBMCs. Data is included for 2 patients only on this matter, and it is important for the overall conclusion of the paper. E.g. a table listing the detection different PBMC vs memory compartment could be useful.

Response: Table S10 includes all data regarding detection differences between PBMC, naïve and memory cells for all patients.

2. Figure 2: Neoantigen reactive T-cells identified for patient 4213 (colon cancer) has been reported by this group previously, thus the current approach doesn't identify any new neoantigen reactive T-cells from an unknown sample. Should be emphasized

Response: We agree with the reviewer and therefore we added the following sentence in page 9:

"We identified a total of two reactivities against neoantigens, both of them also detected in TIL from these patients"

3. Figure 2: For better clarity, authors might consider swapping Figure 2B with 2A. This would establish first that no recognition was observed when bulk PBMCs were stimulated with mutated long peptides.

Response: We understand the rationale behind Reviewer 3 suggestion, however, we prefer to keep the current format.

4. More information should be added on the frequency of the mutations across patient populations. Again a table format could be useful. Arguing the value of these neoepitopes as a shared target would be supported by such information.

Response: We added a table (Table S11) emphasizing the therapeutic potential of targeting KRAS hot-spot mutations.

Reviewers' Comments:

Reviewer #2:

None

Reviewer #3:

Remarks to the Author:

The authors provided the requested information. No further comments